# Effect of Different Compatibilizers on the Mechanical, Flame Retardant, and Rheological Properties of Highly Filled Linear Low-Density Polyethylene/Magnesium Hydroxide Composites

**DOI:** 10.3390/polym15204115

**Published:** 2023-10-17

**Authors:** Beibei Sun, Li Dang, Qiuyan Bi, Rujie Li, Qiuhui Gong, Zhihao Wan, Shiai Xu

**Affiliations:** 1School of Chemical Engineering, Qinghai University, Xining 810016, China; qdhysun@163.com (B.S.); bqy1110@163.com (Q.B.); lrj_superman@163.com (R.L.); 18525842558@163.com (Q.G.); 13525109920@163.com (Z.W.); 2School of Materials Science and Engineering, East China University of Science and Technology, Shanghai 200237, China

**Keywords:** highly filled, linear low-density polyethylene, magnesium hydroxide, compatibilizer, mechanical properties

## Abstract

Maleic anhydride-modified homopolymerized polypropylene (PP-g-MAH) and maleic anhydride-modified polyolefin elastomer (POE-g-MAH) were used as bulking agents to improve the poor processing and mechanical properties of highly filled composites due to high filler content. In this study, a series of linear low-density polyethylene (LLDPE)/magnesium hydroxide (MH) composites were prepared by the melt blending method, and the effects of the compatibilizer on the mechanical properties, flame retardancy, and rheological behavior of the composites were investigated. The addition of the compatibilizer decreased the limiting oxygen index (LOI) values of the composites, but they were all greater than 30.00%, which belonged to the flame retardant grade. Mechanical property tests showed that the addition of the compatibilizer significantly increased the tensile and impact strengths of the LLDPE/60MH (MH addition of 60 wt%) composites. Specifically, the addition of 5 wt% POE-g-MAH increased 154.07% and 415.47% compared to the LLDPE/60MH composites, respectively. The rotational rheology test showed that the addition of the compatibilizer could effectively improve the processing flow properties of the composites. However, due to the hydrocarbon structure of the compatibilizer, its flame retardant properties were adversely affected. This study provides a strategy that can improve the processing and mechanical properties of highly filled composites.

## 1. Introduction

Polyethylene (PE) is one of the most important downstream products of ethylene. It has good chemical stability and considerable low-temperature resistance, making it widely used in many fields such as plastic packaging films, wires and cables, and construction [1,2,3,4,5,6,7]. PE includes low-density polyethylene (LDPE), high-density polyethylene (HDPE), and linear low-density polyethylene (LLDPE) [3,8,9]. LLDPE is a thermoplastic material that is light, non-toxic, and has excellent electrical insulation and chemical resistance. However, the limiting oxygen index (LOI) value of LLDPE is only 17.50%, and the heat resistance is poor, which leads to limited applications in many fields [2,10,11]. The most effective means of solving this problem is the addition of flame retardants.

At present, the halogen and phosphorus flame retardants are used in the market. Although it has a high flame retardant efficiency, the combustion process will produce toxic gases and smoke, causing harmful effects to the human body due to the halogen, nitrogen, phosphorus, and other elements [12,13,14]. Therefore, inorganic flame retardants have attracted great attention. Among them, magnesium hydroxide (MH) is one of the most researched and rapidly developed inorganic flame retardants. It has the advantages of being non-toxic, smokeless, acid-resistant, non-corrosive, having a high decomposition temperature (340–490 °C) (with MgO and H_2_O as decomposition products), and being abundant and inexpensive. However, in practical applications, MH as a flame retardant requires a high loading of 50 wt% to achieve a good flame retardant effect, which severely reduces the mechanical properties of the composites [1,15,16,17,18,19,20,21,22,23]. Therefore, the determination of the flame retardancy of composites is particularly important. The flame retardancy of composites can be evaluated by the determination of LOI and the heat (e.g., THR, pHRR) and smoke (e.g., TSP, pSPR) released from the composites during the combustion process. Among them, the LOI value is the concentration of oxygen in the volume fraction of a substance in a gas mixture of oxygen and nitrogen just able to support its combustion. THR and TSP refer to the total heat and smoke released by the composite material during combustion, respectively. pHRR and pSPR are the maximum of the average value of the heat release rate and the maximum of the average value of the smoke release rate of the composite material during combustion. Wang et al. [17] investigated the flame retardancy of polycarbosilane (PCS)-reinforced polyethylene (PE)/magnesium hydroxide (MNH). The results indicated that PCS-modified MNH was efficient in flame retardant PE. When the content of MNH/PCS was 30 wt%, the LOI value of the PE/MNH/PCS composites was as high as 35.00%. When the content of MNH/PCS was 50 wt%, the peak heat release rate (pHRR) and the total heat release (THR) values of the PE/MNH/PCS composites were reduced by 36.00% and 25.00% compared with PE/MNH, respectively. Jiao et al. [24] reported that when MH was added into ethylene-vinyl acetate (EVA), the LOI values of the composites increased dramatically. Among them, the LOI value of EVA4 was as high as 37.90%. The pHRR, THR, and the peak of the smoke production rate (pSPR) of EVA3 were reduced by 37.60%, 20.70%, and 44.40%, respectively, compared to pure EVA.

To overcome the agglomeration of inorganic fillers in the matrix and to increase the compatibility between inorganic fillers and the matrix, polymer graft copolymers are often used as compatibilizers [25,26,27,28]. For example, in polyolefin composites, the use of polymers with a polar molecular modification such as maleic anhydride grafted polyethylene (PE-g-MAH) is an effective method to improve interfacial adhesion. The polar maleic anhydride part can interact with inorganic fillers through its functional groups, while the long chains of the PE part can interact with the matrix material by forming van der Waals interactions and physical entanglements. Thus, a permanent connection between the inorganic flame retardant additive and the matrix is realized [1]. Shen et al. [29] investigated the effects of two different compatibilizers, maleic anhydride grafted polypropylene (PP-g-MAH) and maleic anhydride grafted polyolefin (POE-g-MAH), on the morphology, thermal stability, and mechanical properties of MH/polypropylene (PP) composites. The results indicated that the introduction of PP-g-MAH or POE-g-MAH into PP/MH composites improved the interfacial interactions between the MH particles and the PP matrix, resulting in a significant increase in the mechanical properties of composites. Hao et al. [30] used maleic anhydride grafted polyethylene to prepare ultra-highly filled wood fiber (WF)/polyethylene composites (UH-WPC). The results showed that ultra-high molecular weight (UHMW) polycarbonate with maleic anhydride grafted polyethylene (MAPE) as matrix or compatibilizer had higher mechanical properties, better creep resistance, and lower water absorption than unexpanded UHMW polycarbonate. The tensile and flexural strengths were increased by 224% and 189%, respectively, with MAPE as a compatibilizer. Liu et al. [31] used 10 wt% maleic anhydride grafted styrene-ethylene-butene-styrene triblock copolymer (SEBS-g-MAH) as a compatibilizer. PP/nano-magnesium hydroxide (nano-MH) composites were prepared by the melt blending method. It was found that SEBS-g-MAH could effectively alpine the dispersion of nano-MH.

In this experiment, highly filled LLDPE/60MH composites were prepared with a MH content of 60 wt%, and the effects of PP-g-MAH and POE-g-MAH on the flame retardant, mechanical, and rheological properties of highly filled LLDPE/60MH composites were investigated. This experiment is or will be a simple and environmentally friendly way for establishing highly filled, environmentally friendly flame retardant composites in industrial production.

## 2. Experimental Procedure

### 2.1. Materials

LLDPE (7042, density = 0.918 g/cm^3^, melt flow index = 5.36 g/10 min (190 °C/5 kg; this value was obtained in laboratory tests)) was supplied by Shanghai Kaibo Species Cable Material Factory Co., Ltd. (Shanghai, China). MH (AR) was supplied by Tianjin Damao Chemical Reagent Factory (Tianjin, China). Maleic anhydride-modified homopolymer polypropylene (PP-g-MAH, grafting rate of 0.50–1.00% density = 0.900 g/cm^3^, melt flow index = 35–70 g/10 min (190 °C/2.16 kg; this value is provided by the merchant)) and maleic anhydride-modified polyolefin elastomers (POE-g-MAH, grafting rate of 0.50–1.00%, density = 0.885 g/cm^3^, melt flow index = 0.3–2.0 g/10 min (190 °C/5 kg; this value is provided by the merchant)) were supplied by Nantong Rizhi High Molecular New Material Technology Co., Ltd. (Nantong China).

### 2.2. Preparation of LLDPE Composites

The LLDPE/MH composites were prepared by the melt blending method. MH, LLDPE, and compatibilizers were premixed at different ratios, and the mixture was added to the Haake torque rheometer at 160 °C and 40 rpm for 10 min to prepare the LLDPE/MH composites. The specific formulations and nomenclatures are shown in Table 1.

### 2.3. Characterization

The LOI value was tested by an oxygen index instrument (JF-6, Nanjing Jionglei Instrument Equipment Co., Ltd., Nanjing, China) according to GB/T 2406-1993 [32] with a specimen of 80 × 10 × 4 mm^3^. Experiments were performed at 23 °C and the average of six samples was reported.

The UL-94 level was tested by a horizontal-vertical burning tester (Zhuhai Huake Testing Equipment Co. Ltd., Zhuhai, China) according to GB/T 2408-2008 [33] with specimen of 125 × 13 × 4 mm^3^. Experiments were performed at room temperature and the average of six samples was reported.

A cone calorimetry test (CCT) was tested by a cone calorimeter (Kunshan Modisco Combustion Technology Instrument Co., Ltd., Kunshan, China) at a heat flux of 35 kW/m^2^ and set an exhaust flow rate of 0.024 m^3^/s according to ISO 5660 [34] with a specimen dimension of 100 × 100 × 3 mm^3^.

The tensile property was measured by a microcomputer-controlled electronic universal testing machine (104B-EX, Shenzhen Vance Test Equipment Co., Ltd., Shenzhen, China) according to GB/T 1040.2-2006 [35], and the specimen was the dumbbell type. The test speed was 10 mm/min and the average of six specimens was reported. The bending properties were measured by a microcomputer-controlled electronic universal testing machine (104B-EX, Shenzhen Vance Testing Equipment Co., Ltd., Shenzhen, China) according to GB/T 9341-2008 [36]. The specimen size was 80 × 10 × 4 mm^3^. The test speed was 2 mm/min and the average of six specimens was reported. The impact properties were measured by an impact tester (501 J-4, Shenzhen Wance Test Equipment Co., Ltd., Shenzhen, China) according to GB/T 1843-2008 [37] with a notched specimen of 80 × 10 × 4 mm^3^ (notch depth of 2 mm) and the energy of the pendulum was 11 J. The average of six specimens was reported.

The composites were quenched and broken in liquid nitrogen, gold-sprayed, and the morphology of the composites was observed on a scanning electron microscope (JSM-7900F, Tokyo, Japan).

The melt index of the composites was tested using a melt index tester (MFI-1322, Chengde Jinjian testing Instrument Co., Ltd., Chengdu, China) at 190 °C and a load of 5 kg according to GB/T 3682 [38]. The average of six specimens was reported.

The equilibrium torque was determined by a torque rheometer (Haake PolyLab QC, Karlsruhe, Germany) at 160 °C and 40 rpm for 10 min.

Dynamic rheological properties were tested by a rotational rheometer (Thermo Scientific Mars40, Waltham, MA, USA) measured at 190 °C with dynamic frequency sweeps from 0.1 to 100 Hz and at 1% strain in a circular specimen with a diameter of 25 mm and a thickness of 1.50 mm.

## 3. Results and Discussion

### 3.1. Flame Retardant Properties of LLDPE and Its Composites

#### 3.1.1. LOI Measurement

Figure 1 shows the results of the LOI values of LLDPE and its composites. From the figure, it can be seen that the LOI value of pure LLDPE was only 19.15%, whereas with the addition of 60 wt% MH, the LOI value of the composites increased to 43.70%, which was an increase of 128.20% as compared to pure LLDPE. It is well known that both PP-g-MAH and POE-g-MAH contain a hydrocarbon skeleton structure, which is easy to burn, such as PP or POE. Therefore, the LOI values of LLDPE/60MH/PP-g-MAH composites decreased slightly with the increase of PP-g-MAH usage as shown in Figure 1. However, the LOI value reached its maximum at 5 wt% of POE-g-MAH. This may be due to the reaction between the polar maleic anhydride groups in POE-g-MAH and the reactive hydroxyl groups on the surface of MH, which improved the dispersion of MH in the LLDPE matrix. However, when the amount of POE-g-MAH was further increased, the LOI values of LLDPE/60MH composites showed a significant decrease, which was attributed to the low grafting rate of maleic anhydride in POE-g-MAH and its own hydrocarbon structure.

#### 3.1.2. UL-94 Test Results

Table 2 summarizes the data from the vertical burning test results for LLDPE and its composites, where t_1_ is the afterflame time of the first ignition, t_2_ is the afterflame time of the second ignition, t_3_ is the afterglow time of the second ignition, and T_f_ is the sum of the residual flame time after the first ignition and the residual flame time after the second ignition, i.e., T_f_ = t_1_ + t_2_. During the experiment, it was found that pure LLDPE was extremely flammable, with ignition of the molten droplets and skimmed cotton observed at the first ignition, and the flame gradually spreading to the fixture until the sample was burned. The UL-94 standard determined the flame retardancy rating to be NR. In contrast, the LLDPE/60MH composite showed better flame resistance, with a residual flame observed only at the bottom of the sample strip after the first ignition and self-extinguishing after 2.59 s of continuous burning. After the second ignition, the residual flame appeared after only 5 s of burning and was completely extinguished after 3.71 s. The LLDPE/60MH composite was judged to have a flame retardant rating of V-0 passing the UL-94 standard. Compared with MH/LLDPE, lint fire was observed when PP-g-MAH or POE-g-MAH was added, but the flame did not spread. The UL-94 rating was downgraded to V-2. It is clear that the vertical combustion rating of the composites decreased after the addition of the compatibilizer, which may be partly due to the addition of the compatibilizer and the flammability of its own structure, and partly due to the fact that the addition of the compatibilizer caused the MH particles to be almost completely encapsulated by the polymer matrix, which generated molten droplets during the combustion process. These results are consistent with the SEM images.

#### 3.1.3. CCT Results

A cone calorimetry test is a comprehensive test to evaluate the combustion performance of composites, which includes the heat release rate (HRR), total smoke produced (TSP), and smoke production rate (SPR), among others [39,40]. Representative data from numerous components were selected and analyzed.

Here, the flammability of LLDPE/60MH, LLDPE/60MH/PP-g-MAH, and LLDPE/60MH/POE-g-MAH composites was evaluated by HRR and THR, and the smoke evacuation capability was evaluated by SPR and TSP. Some other parameters such as time to ignition (T_ign_), pHRR, pSPR, and mass loss rate (MLR) are summarized in Table 3. The variation curves of HRR, THR, SPR, and TSP of LLDPE and its composites are shown in Figure 2. As shown in Figure 2 and Table 3, the pHRR, PSPR, and MLR of pure LLDPE had maximum values among all the composites which were 496.02 kW/m^2^, 0.013 m^2^/s, and 65.16%, respectively. When 60 wt% of MH was added, the pHRR, PSPR, and MLR values of LLDPE/60MH composites had a significant decrease of 250.51 kW/m^2^, 0.010 m^2^/s, and 32.65%, respectively, which were 49.50%, 23.08%, and 49.89%, respectively, compared to pure LLDPE. This indicates that the addition of MH can improve the flame retardant properties of composites.

When the additional amount of PP-g-MAH or POE-g-MAH was small (as the additional amount was 5 wt%), the pHRR and pSPR values of the LLDPE/60MH/PP (or POE)-MAH composites were slightly decreased, and the THR and TSP values were slightly increased compared to the LLDPE/60MH composites, while the values of LLDPE/60MH and LLDPE/60MH and LLDPE/60MH and LLDPE/60MH/PP (or POE)-MAH composites showed little difference in MLR values. This indicates that the small addition of PP-g-MAH and POE-g-MAH has little effect on the flame retardancy of LLDPE/60MH composites, which may be due to the fact that the addition of compatibilizers improves the dispersion of MH particles in the LLDPE matrix, which has a positive effect on the flame retardancy of the composites. In particular, the LLDPE/60MH/5PP-g-MAH composite with the 5 wt% PP-g-MAH addition showed the lowest pHRR and pSPR values. However, when the compatibilizer was added up to 20 wt%, the pHRR and THR values of the composites were significantly larger than those of the LLDPE/60MH composites, which could be attributed to the fact that PP-g-MAH and POE-g-MAH are easily combustible due to their own hydrocarbon structure. It can also be seen from Table 3 that the MLR values of the LLDPE/60MH/PP (or POE)-g-MAH composites were significantly lower than those of the pure LLDPE ones, which suggests that the addition of the compatibilizer helps the composites to form more residual carbon during the combustion process. This is consistent with the results in Figure 3.

### 3.2. Mechanical Properties of LLDPE and Its Composites

Figure 4 shows the mechanical properties of LLDPE and its composites. As can be seen from Figure 4, pure LLDPE had the largest elongation strain at break (748.37%) and high tensile strength (20.75 MPa). With the addition of MH, the elongation at break (5.43%) and tensile strength (8.60 MPa) of LLDPE/60MH composites decreased drastically, which could be attributed to the fact that the composites became more brittle due to the stress concentration effect with the addition of a large amount of MH, which resulted in a significant decrease in the mechanical properties. However, the elongation at break and tensile strength of LLDPE/60MH/PP or (POE)-g-MAH composites were improved to different degrees compared to the LLDPE/60MH composites, which may be attributed to the fact that the flexible long-chain segments of PP-g-MAH and POE-g-MAH can be physically entangled with the long-chain segments of the LLDPE matrix, and the polar MAH groups reacted chemically with the reactive hydroxyl groups on the MH surface. Therefore, the addition of the compatibilizer improved the bonding between MH particles and the LLDPE matrix, and the stronger bonding could better transfer the stress, which led to an increase in the tensile strength of the composites. It is also crucial for the determination of the impact strength of the highly filled polymer matrix composites, which can be utilized to assess the toughness of the composites. As can be seen in Figure 4D, the impact strength of the composites tended to increase with the increase of the POE-g-MAH addition, where the composite with the 5 wt% POE-g-MAH addition had the largest impact strength (82.99 kJ/m^2^) relative to the pure LLDPE (50.84 kJ/m^2^) and LLDPE/60MH composites (16.10 kJ/m^2^) by 63.24% and 415.47%, respectively; this indicates that the toughness of the LLDPE/60MH composites increased with the increase of the POE-g-MAH content. However, with the increase of the PP-g-MAH addition, the impact strength program of the composites tended to decrease, but both were higher than that of the LLDPE/60MH composites. This is consistent with the fracture nominal results in Figure 4B, where POE-g-MAH was based on a polyolefin elastomer, which was added to the LLDPE/60MH composite to play a toughening role. It made the impact properties of LLDPE/60MH/POE-g-MAH better than LLDPE/60MH/PP-g-MAH composites. Figure 4E,F characterize the flexural properties of the composites. It can be seen that the addition of PP-g-MAH can improve the flexural strength and flexural modulus of the composites. The addition of 5 wt% PP-g-MAH composites increased the flexural strength by 169.80% and 9.95% (25.03 MPa, 1550.19 MPa) over pure LLDPE (10.20 MPa) and LLDPE/60MH (25.03 MPa) composites, respectively. The composites with the addition of POE-g-MAH showed an increase in flexural strength and flexural modulus relative to pure LLDPE, but a slight decrease relative to the LLDPE/60MH composites. This is contrary to the fracture nominal results in Figure 4B, where PP-g-MAH was based on homopolymerized polypropylene, which was added to the LLDPE/60MH composite to provide some reinforcement. Thus, the LLDPE/60/MH/PP-g-MAH flexural properties were superior to those of LLDPE/60/MH/POE-g-MAH composites. The improvement in the mechanical properties of LLDPE/60MH composites stemmed from the improved dispersion of MH by the addition of the compatibilizer (in agreement with the results of SEM tests in Figure 5). This mechanical result indicates that PP-g-MAH and POE-g-MAH are not only good compatibilizers in LLDPE/60MH composites, but also have a certain reinforcing and toughening effect on LLDPE/60MH composites.

### 3.3. Micro-Interfacial Observation

Figure 5 shows the SEM images of frozen sections of LLDPE/60MH, LLDPE/60MH/5PP-g-MAH, and LLDPE/60MH/5POE-g-MAH composites. As can be seen from Figure 5A–C, in the LLDPE/60MH composites, the MH surface was smoother and less bonded to the LLDPE matrix, and the MH was stacked together and agglomerated more severely, leading to their poor compatibility with the LLDPE matrix. This is consistent with the results of the mechanical properties test. As shown in Figure 5D–I, the MH was uniformly dispersed in the LLDPE matrix after the addition of PP-g-MAH or POE-g-MAH, the MH in (F), and (I) were not stacked together, but embedded in the LLDPE matrix, forming a similar “honeycomb”. The agglomeration phenomenon was weakened, and the dispersion was improved. This indicates that PP-g-MAH and POE-g-MAH played a good compatibility role.

### 3.4. MFI Test

The melt flow index (MFI) can be used to characterize the processing flow properties of composites [41]. As shown in Figure 6, the MFI of pure LLDPE was 5.36 g/10 min, and the MFI of the composites decreased to 0.38 g/10 min with the addition of 60 wt% MH. The addition of PP-g-MAH with a high melt flow index (MFI = 46.26 g/10 min) to the composites increased the MFI value as compared to LLDPE/60MH composites. The addition of POE-g-MAH with a low melt flow index (MFI = 0.75 g/10 min) to the composites observed less outflow from the composites and a sharp decrease in their MFI values. The flowability of the highly filled LLDPE/MH composites was significantly increased by the addition of PP-g-MAH, and the higher the content, the higher the MFI value and the better the flowability of the composites, which may be attributed to the fact that PP-g-MAH itself had better flowability, and its addition to the matrix reduced the agglomeration of the MH and improved the dispersion of the MH particles. The addition of POE-g-MAH did not improve the flow of the highly filled LLDPE/MH composites, probably due to the low MFI value of POE-g-MAH itself.

### 3.5. Rheological Properties of LLDPE and Its Composites

#### 3.5.1. Torque Analysis

Figure 7 shows the torque versus time plot for LLDPE and its composites. As shown in the figure, pure LLDPE had the lowest equilibrium torque value of 4.72 N·m and the composite with 60 wt% of MH added had a torque value of 5.24 N·m, which was an increase of 11.02% compared to pure LLDPE. With the increase of the PP-g-MAH addition, the equilibrium torque value of the LLDPE/60MH/PP-g-MAH composites decreased, and the processing flow performance was improved; with the increase of the POE-g-MAH addition, the equilibrium torque value of the LLDPE/60MH/POE-g-MAH composites increased. The equilibrium torque value of the LLDPE/60MH/PP-g-MAH composites was significantly lower than that of pure LLDPE. The equilibrium torque values of the composites were significantly lower than those of the LLDPE/60MH/POE-g-MAH composites (in agreement with the results of melt flow rate measurements). This can be attributed to the higher MFI values of PP-g-MAH than POE-g-MAH.

#### 3.5.2. Dynamic Rheological Properties

Rheological properties are important for characterizing the microstructure of composites because the viscoelastic response of a substance is related to both the short-range structure of the filler and the long-range interaction of the filler with the matrix [41,42,43,44]. Figure 8A illustrates the complex viscosity of LLDPE and its composites at different angular frequencies. The complex viscosity *η** of the pure LLDPE sample remained almost constant at low angular frequencies, but decreased with further increases in angular frequency ω. After the addition of MH particles, the *η** of LLDPE/60MH composites increased significantly, which indicates that a large number of MH particles can hinder the movement of LLDPE molecular chains, leading to a sharp increase in *η**. The *η** of LLDPE/60MH composites decreased significantly with increasing ω. The Newtonian flat zone of the melts disappeared, and the composites exhibited pseudo-plastic fluid (non-Newtonian) properties in the full range of the angular frequency, i.e., typical shear-thinning behavior [22]. With the addition of the compatibilizer, the decreasing trend of the composite sample curves was the same, indicating that the addition of the compatibilizer does not change the shear-thinning behavior of the LLDPE/60MH composites.

*G*′ mainly responds to the elastic properties of the material microstructure, while *G*″ mainly responds to the intermolecular interactions and free migration of molecular chains in the melt [42,45,46]. The loss modulus *G*′ with angular frequency ω are shown in Figure 8B,C, respectively. The double logarithmic dependence of *G*′ and *G*″ on ω for pure LLDPE in the low-frequency end region was consistent with the classical linear viscoelasticity theory, i.e., *G*′ ∝ *ω*^2^ and *G*″ ∝ *ω* [22,47], which suggests that the molecular chains of LLDPE can be completely relaxed in the low-frequency region. The incorporation of MH particles modified the viscoelasticity of LLDPE, and the moduli of LLDPE/60MH composites were significantly increased and no end relaxation was observed in the low-frequency region. No end relaxation was observed in the low-frequency region, which implied that the relaxation process of LLDPE molecular chains in LLDPE/60MH composites will take place at lower frequencies and also take a longer time [41,48]. With the addition of compatibilizers, the modulus of the composites decreased significantly, which was mainly due to the fact that the compatibilizers acted as a dispersing agent for the MH particles, weakening the interaction between the MH particles and reducing the modulus of the samples.

## 4. Conclusions

In this study, a series of LLDPE/60MH composites were prepared by the melt blending method. The effects of PP-g-MAH and POE-g-MAH as compatibilizers on the mechanical properties, flame retardancy, and rheological behavior of the LLDPE/60MH composites were investigated. The results showed that the LOI values of the composites were still greater than 30.00% after the addition of the compatibilizers and belonged to the flame retardant level, in which the LOI value of the LLDPE/60MH/5POE-g-AMH composite (49.30%) was 12.80% higher than that of the LLDPE/60MH composite (43.70%). The experimental results show that the addition of PP-g-MAH can improve the processing fluidity of LLDPE/60MH composites and enhance the mechanical properties of the composites, and the addition of POE-g-MAH can confer certain toughness to LLDPE/60MH composites, which increases the nominal strain at break and impact strength. However, due to the hydrocarbon structure of PP-g-MAH and POE-g-MAH, their addition may affect the flame retardancy of the composites.

## Figures and Tables

**Figure 1 polymers-15-04115-f001:**
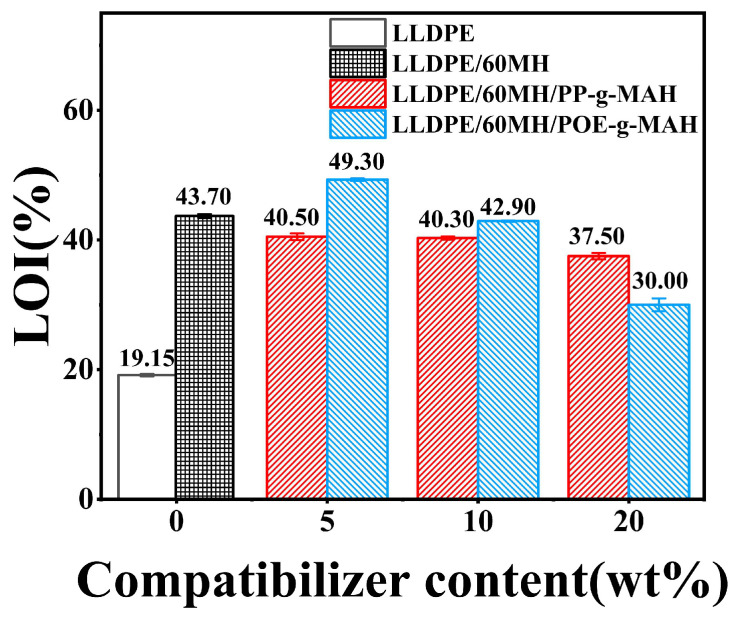
The LOI values for LLDPE and its composites.

**Figure 2 polymers-15-04115-f002:**
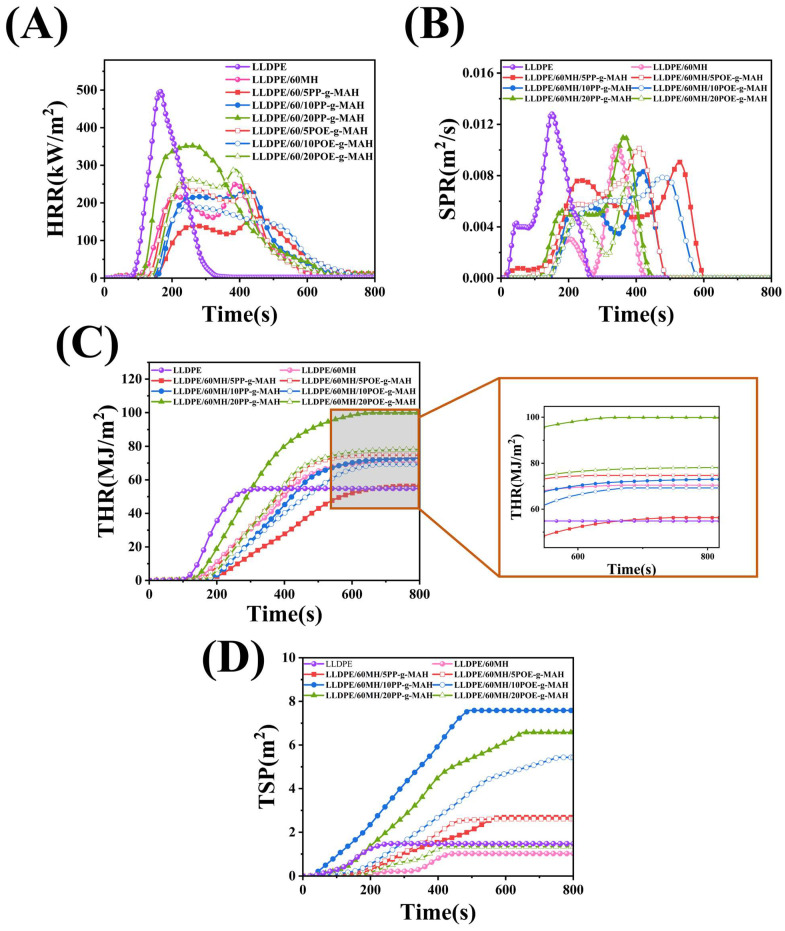
Cone combustion parameters of LLDPE and its composites: (**A**) HRR, (**B**) SPR, (**C**) THR, and (**D**) TSP.

**Figure 3 polymers-15-04115-f003:**
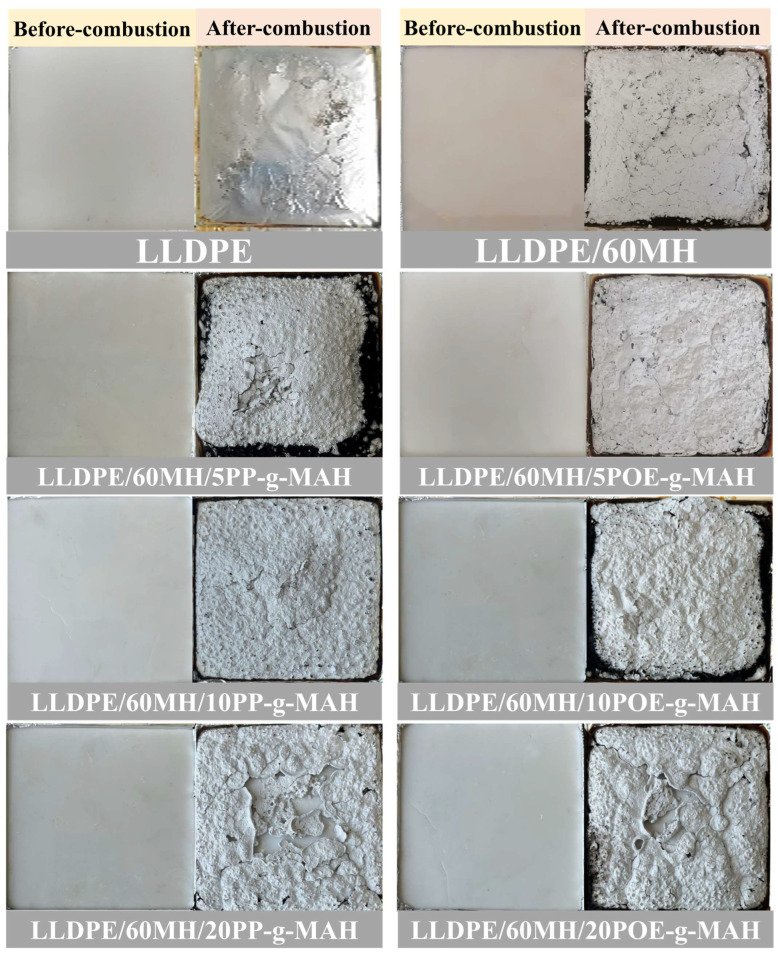
Comparative photographs of LLDPE and its composites before-combustion and after-combustion.

**Figure 4 polymers-15-04115-f004:**
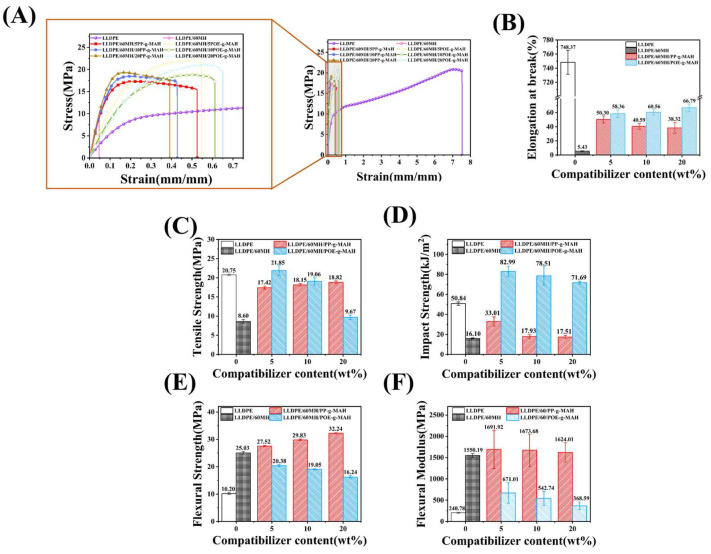
Mechanical properties of LLDPE and its composites: (**A**) tensile stress–strain curves, (**B**) elongation at break, (**C**) tensile strength, (**D**) impact strength, (**E**) flexural strength, and (**F**) flexural modulus.

**Figure 5 polymers-15-04115-f005:**
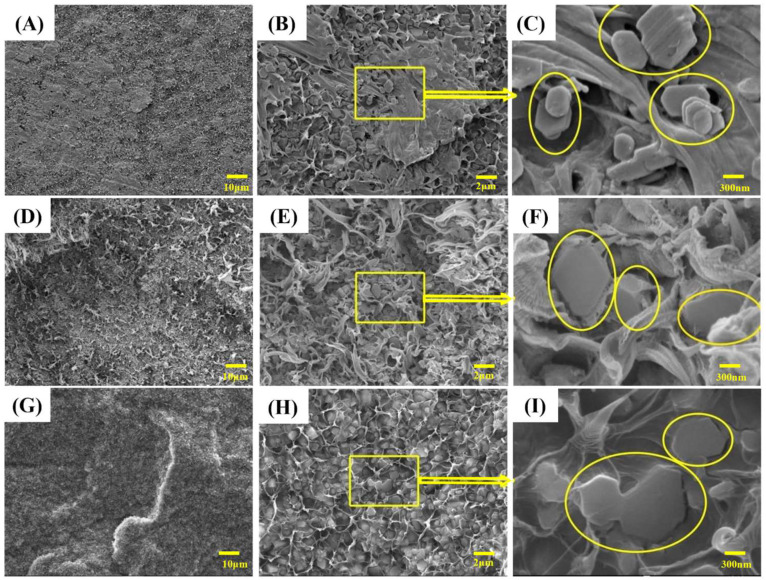
SEM images of quenched cross-sections of LLDPE and its composites: (**A**–**C**) are LLDPE/60MH composites; (**D**–**F**) are LLDPE/60MH/5PP-g-MAH composites; (**G**–**I**) are LLDPE/60MH/5POE-g-MAH composites.

**Figure 6 polymers-15-04115-f006:**
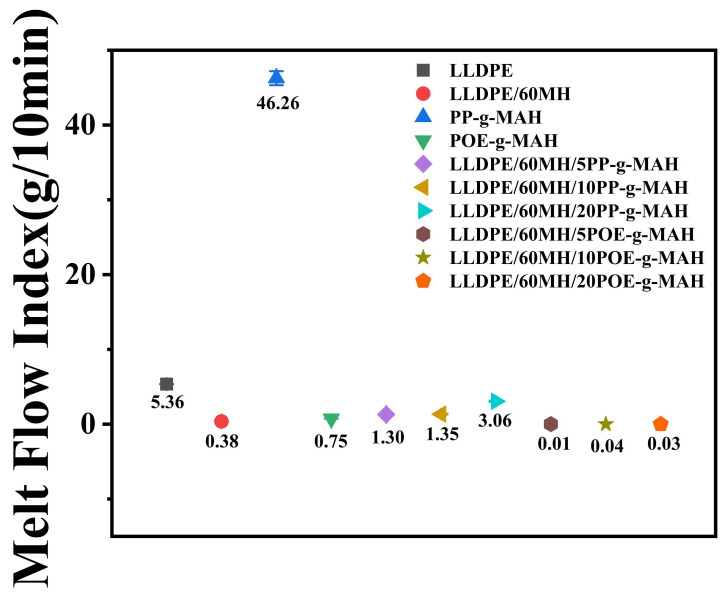
Melt flow index graph for LLDPE and its composites.

**Figure 7 polymers-15-04115-f007:**
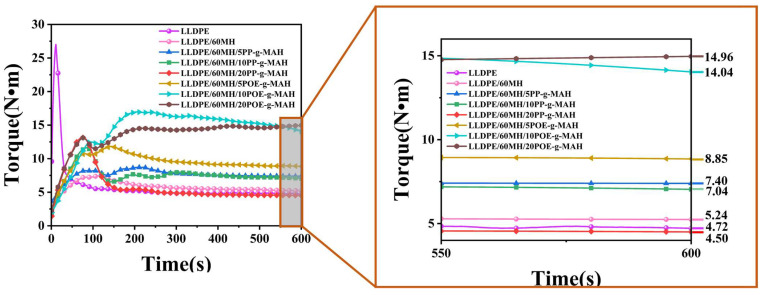
Torque–time curves for LLDPE and its composites.

**Figure 8 polymers-15-04115-f008:**
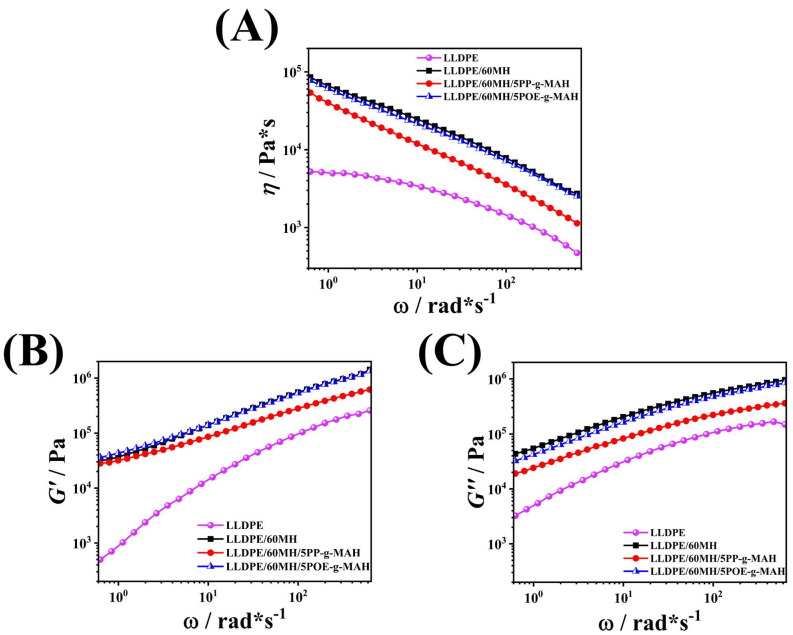
Plot of complex viscosity *η** (**A**); energy storage modulus *G*′ (**B**); loss modulus *G*″ (**C**) versus angular frequency *ω* for LLDPE and its composites.

**Table 1 polymers-15-04115-t001:** The formulations and nomenclatures of LLDPE/MH composites.

Samples	LLDPE	MH	PP-g-MAH	POE-g-MAH
LLDPE	100	0	0	0
LLDPE/60MH	40	60	0	0
LLDPE/60MH/5PP-g-MAH	35	60	5	0
LLDPE/60MH/10PP-g-MAH	30	60	10	0
LLDPE/60MH/20PP-g-MAH	20	60	20	0
LLDPE/60MH/5POE-g-MAH	35	60	0	5
LLDPE/60MH/10POE-g-MAH	30	60	0	10
LLDPE/60MH/20POE-gMAH	20	60	0	20

**Table 2 polymers-15-04115-t002:** LLDPE and its composites’ vertical combustion test records.

Samples	t_1_/s	t_2_/s	T_f_/s	t_3_/s	(t_2_ + t_3_)/s	Spreading to Fixtures	Cotton Ignition	UL-94 Rating
LLDPE	320.00	0	0	0	0	Y	Y	NR
LLDPE/60MH	2.25	4.59	6.83	3.71	8.30	N	N	V-0
LLDPE/60MH/5PP-g-MAH	2.61	68.29	70.90	12.56	80.85	N	Y	V-2
LLDPE/60MH/10PP-g-MAH	20.46	4.96	25.42	3.05	8.01	N	Y	V-2
LLDPE/60MH/20PP-g-MAH	23.48	3.61	27.09	0	3.61	N	Y	V-2
LLDPE/60MH/5POE-g-MAH	2.54	31.98	34.52	0	31.98	N	Y	V-2
LLDPE/60MH/10POE-g-MAH	21.11	6.30	27.40	0	6.30	N	Y	V-2
LLDPE/60MH/20POE-g-MAH	13.33	2.33	15.65	0	2.33	N	Y	V-2

N represents NO; Y represents YES; NR indicates that no level has been reached.

**Table 3 polymers-15-04115-t003:** Cone calorimetry test data for LLDPE and its composites.

Samples	T_ign_ (s)	pHRR (kW/m^2^)	THR (MJ/m^2^)	pSPR (m^2^/s)	TSP (m^2^)	MLR (%)
LLDPE	75	496.02	54.84	0.013	1.48	65.16
LLDPE/60MH	107	250.51	70.45	0.010	1.03	32.65
LLDPE/60MH/5PP-g-MAH	157	160.67	56.31	0.009	2.68	32.76
LLDPE/60MH/10PP-g-MAH	161	230.64	72.99	0.008	7.58	41.69
LLDPE/60MH/20PP-g-MAH	112	352.16	99.85	0.011	6.58	46.89
LLDPE/60MH/5POE-g-MAH	134	249.73	74.71	0.010	2.61	38.01
LLDPE/60MH/10POE-g-MAH	152	190.72	69.25	0.008	5.43	40.00
LLDPE/60MH/20POE-g-MAH	146	289.33	78.15	0.007	1.34	45.69

## Data Availability

The data presented in this study are available on request from the corresponding author.

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
