# Peer review of "Effect of Different Compatibilizers on the Mechanical, Flame Retardant, and Rheological Properties of Highly Filled Linear Low-Density Polyethylene/Magnesium Hydroxide Composites"

_polymers, 2023, doi:10.3390/polym15204115_

Round 1
Reviewer 1 Report
The authors developed composites with possible flame retardant potential. The manuscript needs revisions before publication:
> Abstract. “The mechanical properties test showed that after adding the compatibilizer, the composites' tensile strength and impact strength of LLDPE/60MH composites were significantly improved.” Please indicate the % gain for each property mentioned;
Conclude the abstract reporting the main discovery of the manuscript and potential application;
> Introduction. Authors must report the novelty of the manuscript in the introduction, including presenting the gaps in the literature. How important is this investigation? Inform the technological, industrial importance, etc.
> Materials. Authors should add the melt flow index (MFI) of LLDPE, PP-g-MAH, and POE-g-MAH. Please provide the name of the POE-g-MAH coupling agent. Additionally, is it rubber-based?
> Characterization. LOI. What is the temperature and relative humidity of the air during the test?
Cone calorimetry test (CCT) - What is the duct flow rate (m3/s) during testing?
Tensile and flexural property. Inform the speed of the test and the load used. Impact test reporting the value of the pendulum used;
> Figure 1. Was the LOI test based on a single sample? The experimental error must be reported;
Why did the authors not present the LLDPE results for comparative purposes? I suggest reporting to show the effect of the filler used in the polymer matrix;
When increasing the content of PP-g-MA and POE-g-MA, there was an increase in the amount of maleic anhydride. What is the role of maleic anhydride in LOI? It must be discussed in the manuscript;
The results must be compared with the standard adopted by the authors. Are composites easily flammable? The standard referring to the oxygen index (LOI) classifies composites as combustible materials?
> UL-94 test results. Please report the meaning of t1, t2, t3, tf;
> Mechanical properties of LLDPE and its composites
Figure 4(a). I suggest the authors modify the colors to distinguish the PP-g-MA and POE-g-MA composites;
The discussion of mechanical properties needs to be improved. Explain the reason for the difference between PP-g-MA and POE-g-MA. Again, is POE elastomeric? How this affected mechanical performance, especially impact strength and elongation at break;
Does the stress vs strain curve in Figure 4 (a) refer to the tensile or flexural test?
The mechanical results of LLPDE would be important, thus better understanding the effect of filler and coupling agents;
Minor editing of English language required
Reviewer 2 Report
The article by Sun B. et al. investigates the flame retardancy of polyethylene containing 60% magnesium hydroxide in the presence of two different grafted copolymers as compatibilizers. The authors show how the addition of different concentrations of compatibilizers changes the flame retardancy of filled polyethylene and also affects its linear viscoelasticity and strength properties. The article contains several shortcomings that should be eliminated before publication.
Specific comments are as follows.
The lines are not numbered, which complicates the review of the article.
Abstract: “LOI tests”, “LLDPE/60MH”. Abbreviations should be deciphered when first used.
Introduction: “limiting oxygen index (LOI)”, “the peak heat release rate (pHRR) and the total heat release (THR)”, “the smoke production rate”. The definition (physical meaning) of these quantities and their importance as parameters for assessing flame retardancy should be given.
Page 2. “In order to improve the agglomeration”. The authors probably had the following in mind: “In order to improve the disagglomeration”.
Materials: “LLDPE (7042)”. The authors should provide the molecular weight, dispersity, and melt flow index of polyethylene.
Materials: “PP-g-MAH”, “POE-g-MAH”. The authors should provide the molecular weight of these polymers. In addition, it should be specified what the authors specifically mean by the word "polyolefin" (POE), since polyolefins are a class of polymers including a dozen different polymers.
Characterization: “at 1% strain”. This is a large strain for highly filled samples. The authors should give the value of the critical strain limiting the linear viscoelastic region for their filled samples.
Characterization: "The morphology of frozen section". It follows from the description that the authors did not apply metal deposition to the surface of the samples. If it is really so, it should be specified.
Figure 1. There is a lack of data for pure LLDPE, which should be provided, at least in words in the text of the paper.
Tables 2 and 3. Data for pure LLDPE should be reported.
Page 5, Figure 4: “which may be attributed to the fact that the composites become more brittle after the addition of a large amount of MH due to the stress concentration effect, resulting in a significant decrease in the mechanical properties”. The authors do not provide data for pure polyethylene to make this conclusion.
Figure 4. Data for pure LLDPE should be provided.
Figure 6. Legends contain a new designation for samples, which has not been used before. The authors should use the same designation of samples throughout the text. In addition, Figure 6 should contain data for pure polyethylene.
Page 10: “the larger the complex viscosity, the poorer the fluidity of the polymer melt, i.e., the poorer the polymer's processing flow properties”. This is an incorrect statement. The complex viscosity is not equal to the steady-state viscosity for highly filled composites and therefore has nothing in common with the flow of the composites (their complex viscosity may be 10 times higher than their steady-state viscosity).
Page 10: “Since LLDPE/60MH composites contain a large amount of MH, which hinders the movement of LLDPE molecular chains”. This is not an accurate description. A large amount of filler leads to the formation of a percolation network in polyethylene, giving it a yield stress due to which the viscosity tends to infinity with a decrease in angular frequency. The presence of percolation structure is indicated by the constant storage modulus at low frequencies in Fig. 6 (see, e.g., 10.1007/s00397-014-0770-6, where polyethylene containing large amounts of boron hydroxide in the form of metaboric acid as a filler exhibits similar behavior).
Page 10: “and can be regarded as the entropic elasticity of the molecular chain segments”. This interpretation is suitable only for pure polymer. For a highly filled polymer, the elasticity of the percolation network of filler particles (see the above reference) is added to the elasticity of the polymer chains, which should be noted.
Conclusions. “the addition of PP-g-MAH or POE-g-MAH improved the processing fluidity of the composites”. The authors did not measure the fluidity of the samples, i.e., the dependence of their steady-state viscosity on shear rate.
The English language requires moderate editing.
Round 2
Reviewer 1 Report
The authors responded clearly to the questions. In addition, the recommendations were accommodated in the revised manuscript.
Minor editing of English language required
Reviewer 2 Report
The authors have substantially improved the manuscript for its publication.
The English language requires moderate editing.